# Asymmetric Synthesis of Three Alkenyl Epoxides: Crafting the Sex Pheromones of the Elm Spanworm and the Painted Apple Moth

**DOI:** 10.3390/molecules29092136

**Published:** 2024-05-04

**Authors:** Yun Zhou, Jianan Wang, Beijing Tian, Yanwei Zhu, Yujuan Zhang, Jinlong Han, Jiangchun Zhong, Chenggang Shan

**Affiliations:** 1Institute of Industrial Crops, Shandong Academy of Agricultural Sciences, Jinan 250100, China; zysass2021@163.com (Y.Z.); tianbeijing07@126.com (B.T.); zhuyanwei96@163.com (Y.Z.); zhangyujuan@caas.cn (Y.Z.); goldendragonh@163.com (J.H.); 2Department of Applied Chemistry, China Agricultural University, Beijing 100193, China; xxjnwang@163.com (J.W.); zhong@cau.edu.cn (J.Z.)

**Keywords:** sex pheromone, alkenyl epoxides, elm spanworm, painted apple moth, asymmetric synthesis

## Abstract

A concise synthesis of the sex pheromones of elm spanworm as well as painted apple moth has been achieved. The key steps were the alkylation of acetylide ion, Sharpless asymmetric epoxidation and Brown’s P2-Ni reduction. This approach provided the sex pheromone of the elm spanworm (**1**) in 31% total yield and those of the painted apple moth (**2**, **3**) in 26% and 32% total yields. The ee values of three final products were up to 99%. The synthesized pheromones hold promising potential for use in the management and control of these pests.

## 1. Introduction

The elm spanworm and the painted apple moth, as lepidopteran pests, have inflicted considerable economic damage upon both forestry and horticulture [1]. The elm spanworm, *Ennomos subsignaria* (Hübner, Kassel, Germany), is distributed in Canada, the eastern region of the United States and Newfoundland and mainly damages deciduous tree species, such as hickory (*Caryra*), ash (*Franxinus*), oak (*Quercus*), red maple (*Acer rubrum*), elm (*Ulmus*), basswood (*Tilia*), beech (*Fagus*), horse chestnut (*Aesculus*) and so on [2,3,4]. The painted apple moth, *Teia anartoides* (Walker), is a pestilential species prevalent in Australia and New Zealand, whose larvae feed on a variety of plants [5,6]. It is a severe threat to forested areas as well as to agricultural, horticultural and silvicultural crops. Presently, the management of these two pest species mainly relies on the application of chemical pesticides [7,8]. As conflicts between agricultural production and environmental matters become more acute, developing green and efficient solutions to pest management gains more importance.

Pest management based on pheromones is an eco-friendly, high effective and promising solution [9]. Significantly, chiral alkenyl epoxides are extensively identified within the pheromonal components and sex attractants of various lepidopteran families. These sex pheromones, comprising unsaturated hydrocarbons and chiral epoxides, are deemed crucial for their biological activity and for facilitating species-specific identification [1]. In 2010, Silk and coworkers first recognized (2*S*,3*R*)-2-((*Z*)-oct-2-enyl)-3-nonyloxirane (**1**) as the sex pheromone of elm spanworm by means of field study and GC–MS coupled with a GC/EAD test [3]. (2*S*,3*R*)-2-((*Z*)-oct-2-enyl)-3-decyloxirane (**2**) and (2*S*,3*R*)-2-((*Z*)-oct-2-enyl)-3-undecyloxirane (**3**) were successfully determined to be the electroantennographically (EAG) active components of the pheromone of the painted apple moth, female *Teia anartoides*, by El-Sayed’s team in 2005 [5,6]. Notably, (2*S*,3*R*)-2-((*Z*)-oct-2-enyl)-3-undecyloxirane (**3**) was also found to be the minor component of the sex pheromone emitted by the females of the ruby tiger moth, *Phragmatobia fuliginosa*, in 1986 [10]. Considering the significant role of these pheromones in pest biological control, there have been several reports on their synthetic methods, and the reported routes are mainly based on chiral pools and asymmetric synthesis [4,10,11,12,13,14]. Employing the chiral pool strategy, Pougny’s team reported the synthesis of compound **3** from d-xylose as a chiral precursor in 10 steps, achieving an approximate total yield of 25% in 1986 [10]. In 1992, Mori and coworkers realized the synthesis of compound **2** with a total yield of 33% in 6 steps and **3** with a total yield of 47% by utilizing Amano’s lipase PS-C kinetic separation of racemic epoxides to construct chiral starting materials [13]. In 2011, MaGee’s group achieved the synthesis of alkenyl epoxide **1** in 14 steps using methyl tartrate as a chiral template, with a total yield of 13% and an enantiopurity of 93.4% [4]. As for constructing chiral centers by means of asymmetric synthesis, Mori’s team synthesized **3** in 5 steps through Sharpless asymmetric oxidations with a total yield of 14.0% and an enantiopurity of 99% in 1989 [15]. In 1991, Millar’s group devised the synthesis of **2** and **3** in 6 steps, with a total yield of approximately 25% and an optical purity of 88% [14]. In 2011, MaGee and coworkers synthesized alkenyl epoxide **1** through 12 steps involving Sharpless asymmetric dihydroxylation, with a total yield of 12.8% and an enantiopurity of 99% [4]. There are still some drawbacks in reported approaches, such as the use of expensive and uncommon starting materials, the low optical purity of final product, long synthesis steps and low total yield. Therefore, developing a short and efficient approach with inexpensive starting materials, nontoxic reagents and a high enantiopurity of final product are still demanding. Herein, we realized the efficient asymmetric synthesis of the sex pheromone component of elm spanworm *Ennomus subsignaria* (Hübner), (2*S*,3*R*)-2-((*Z*)-oct-2′-enyl)-3-nonyl oxirane (**1**), as well as those of painted apple moth, *Teia anartoides* (Walker), (2*S*,3*R*)-2-((*Z*)-oct-2′-enyl)-3-decyl oxirane (**2**) and (2*S*,3*R*)-2-((*Z*)-oct-2′-enyl)-3-undecyl oxirane (**3**) (Figure 1). Our synthesis was conducted efficiently, yielding the target pheromone with high overall yield and exceptional enantiomeric purity.

## 2. Results and Discussion

### 2.1. Retrosynthetic Analysis

A retrosynthetic analysis of the sex pheromone **1** is illustrated in Figure 1. (2*S*,3*R*)-2-((*Z*)-Oct-2-enyl)-3-nonyloxirane (**1**) could be prepared via a *cis*-selective reduction of (2*S*,3*R*)-2-(oct-2-ynyl)-3-nonylepoxyethane (**18**). The nucleophilic substitution between alkyne **17** and the triflate derived from **14** could afford epoxy alkynyl alcohol **18**. The chiral block **14** was envisaged to be synthesized via the Sharpless asymmetric epoxidation of *cis*-allylic alcohol **11**, which could be obtained from the alkylation of propargyl alcohol (**7**) with 1-bromononane (**4**) and the subsequent Brown’s P2-Ni reduction. Obviously, the sex pheromones **2** and **3** could be prepared following a similar approach to sex pheromone **1**.

### 2.2. Synthesis of Chiral Epoxy Alcohols

Our synthesis began with the preparation of chiral epoxy alcohols **14**–**16**, as depicted in Figure 2. According to the procedure reported by Dallavalle [16], the alkylation of propargyl alcohol **7** with 1-bromononane (**4**) provided dodec-2-yn-1-ol (**8**) in excellent yield. Subsequently, the pure *cis*-selective product (**11**) was generated (95% yield, *Z*:*E* > 99:1) through the reduction of the alkyne bond in **8** with Brown’s catalyst P2-Ni [17,18]. The next process was the crucial step in this transformation, which introduced chiral epoxy alcohol to primary allylic alcohol **11** via asymmetric Sharpless epoxidation. The synthesis of ((2*S*,3*R*)-3-nonyloxiran-2-yl) methanol (**14**) was accomplished (82% yield, 88% ee), catalyzed by L-(+)-diisopropyltartrate and Ti(O*i*Pr)_4_ [19,20]. After one recrystallization using petroleum ether, the ee value of **14** increased to over 99%, which was determined via the ^1^H NMR analysis of the Mosher ester of **14** [20,21]. Similarly, using primary bromides **5** and **6** as starting materials, respectively, desired chiral blocks **15** and **16** could be obtained through the above-mentioned synthesis process of **14**. The enantiomeric purity of **15** and **16** could also be improved to up to 99% after recrystallization with petroleum ether.

### 2.3. Synthesis of Target Compounds

With chiral epoxy alcohols in hand, we focused on the synthesis of the target pheromones **1**–**3**, as outlined in Figure 3. The hydroxyl group in **14** was activated by treating with trifluoromethanesulfonic anhydride to afford a corresponding triflate. Then, the alkynylation coupling reaction between the in situ-generated triflate and **17** was undertaken, offering the desired epoxy alkyne **18** in 80% yield [22]. Following a similar procedure, corresponding epoxy alkynes **19** and **20** were afforded with good yields. Finally, the *cis*-selective reduction of **18**, **19** and **20** produced the corresponding target compounds **1** (90% yield, *Z*:*E* > 99:1), **2** (88% yield, *Z*:*E* > 99:1) and **3** (92% yield, *Z*:*E* > 99:1), respectively [17,23]. It is noteworthy that we employed the alkynylation/reduction approach to synthesize compounds **14**–**16** into target pheromones **1**–**3**, achieving conversions with yields ranging from 69% to 75% in two steps. This approach has demonstrated high efficiency and straightforward execution. The structures of these three sex pheromones were characterized by ^1^HNMR, ^13^CNMR, HRMS and specific rotation (^1^HNMR and ^13^CNMR spectras are available in the Appendix A), which were consistent with the references [4,5].

## 3. Materials and Methods

### 3.1. General Information

All reactions were carried out within a Schlenk line system under an inert atmosphere of argon. Commercially available reagents were utilized as received, without additional purification. In contrast, solvents underwent distillation following standard procedures prior to use. Column chromatography was generally performed on a silica gel (200–300 mesh), and elution was performed with petroleum ether and ethyl acetate. ^1^H and ^13^C NMR spectra were recorded on a Bruker DP-X500 MHz spectrometer (Bruker Corporation, Beijing, China). Chemical shifts were reported in ppm relative to internal tetramethylsilane for ^1^H NMR and CDCl_3_ (77.16 ppm) for ^13^C NMR. High-resolution mass spectra (HRMS) were collected on Waters LCT Premier™ with an ESI mass spectrometer (Waters Corporation, Beijing, China). Optical rotations were determined by a Rudolph Research Analytical AUTOPOL-IV. Melting points were measured on a STUART-SMP3 Melt-Temp apparatus without correction (Stuart Equipment, Beijing, China).

### 3.2. Synthesis of Dodec-2-yn-1-ol (***8***)

Under an argon atmosphere, *n*-BuLi (23.33 mL, 2.4 M in *n*-hexane, 56 mmol, 4.0 equiv.) was added slowly to a stirred solution of prop-2-yn-1-ol **7** (1.57 g, 28 mmol, 2.0 equiv.) and HMPA (15.05 g, 84 mmol, 6.0 equiv.) in THF (25 mL) at −78 °C. The resulting mixture was allowed to warm to −30 °C and stirred for 3 h. Then, 1-bromononane **4** (2.90 g, 14 mmol, 1.0 equiv.) was added to the reaction mixture dropwise and stirred for 30 min at −30 °C. After the reaction mixture was warmed to room temperature and stirred overnight, it was quenched with saturated aqueous NH_4_Cl (30 mL). The organic phase was separated, and the aqueous phase was extracted with diethyl ether (3 × 50 mL). The combined organic phases were washed with brine, dried over anhydrous Na_2_SO_4_, and concentrated under reduced pressure to obtain crude product. The crude product was purified via silica gel chromatography (petroleum ether/ethyl acetate 10:1) to afford dodec-2-yn-1-ol (**8**) (2.36 g, 93% yield) as a pale yellow oil. ^1^H NMR (500 MHz, CDCl_3_) δ 4.25 (t, *J* = 2.0 Hz, 2H), 2.20 (tt, *J* = 7.0, 2.0 Hz, 2H), 1.60 (br s, 1H), 1.53–1.47 (m, 2H), 1.38–1.26 (m, 2H), 1.31–1.26 (m, 10H), 0.88 (t, *J* = 7.0 Hz, 3H). ^13^C NMR (125 MHz, CDCl_3_) δ 86.83, 78.39, 51.58, 32.01, 29.61, 29.42, 29.28, 29.01, 28.75, 22.81, 18.87, 14.24. HRMS (ESI, *m*/*z*): calculated for [M + H]^+^ C_12_H_23_O 183.1743, found: 183.1754.

### 3.3. Synthesis of Tridec-2-yn-1-ol (***9***)

Using a similar procedure as for alkynyl alcohol **8**, 1-bromodecane **5** (3.10 g, 14 mmol, 1.0 equiv.) and prop-2-yn-1-ol (**7**) (1.57 g, 28 mmol, 2.0 equiv.) afforded tridec-2-yn-1-ol (**9**) (2.54 g, 92% yield) as a pale yellow oil. ^1^H NMR (500 MHz, CDCl_3_) δ 4.25 (t, *J* = 2.0 Hz, 2H), 2.21 (tt, *J* = 7.0, 2.0 Hz, 2H), 1.59 (br s, 1H), 1.53–1.47 (m, 2H), 1.38–1.34 (m, 2H), 1.31–1.27 (m, 12H), 0.88 (t, *J* = 7.0 Hz, 3H). ^13^C NMR (125 MHz, CDCl_3_) δ 86.82, 78.39, 51.58, 32.04, 29.72, 29.66, 29.46, 29.28, 29.02, 28.75, 22.82, 18.87, 14.25. HRMS (ESI, *m*/*z*): calculated for [M + H]^+^ C_13_H_25_O 197.1900, found: 197.1910.

### 3.4. Synthesis of Tetradec-2-yn-1-ol (***10***)

Using a similar procedure as for the synthesis of **8**, 1-bromoundecane **6** (3.30 g, 14 mmol, 1.0 equiv.) and prop-2-yn-1-ol **7** (1.57 g, 28 mmol, 2.0 equiv.) afforded tetradec-2-yn-1-ol (**10**) (2.65 g, 90% yield) as a light yellow oil. ^1^H NMR (500 MHz, CDCl_3_) δ 4.25 (s, 2H), 2.21 (tt, *J* = 7.0, 2.0 Hz, 2H), 1.53–1.47 (m, 3H), 1.38–1.34 (m, 2H), 1.31–1.26 (m, 14H), 0.88 (t, *J* = 7.0 Hz, 3H). ^13^C NMR (125 MHz, CDCl_3_) δ 86.86, 78.38, 51.61, 32.06, 29.78, 29.76, 29.67, 29.49, 29.29, 29.03, 28.75, 22.83, 18.88, 14.26. HRMS (ESI, *m*/*z*): calculated for [M + H]^+^ C_14_H_27_O 211.2056, found: 211.2062.

### 3.5. Synthesis of (Z)-Dodec-2-en-1-ol (***11***)

Under a hydrogen atmosphere, a suspension of NaBH_4_ (0.38 g, 10.0 mmol, 1.0 equiv.) in ethanol (10 mL) was added slowly to a stirred solution of Ni(OAc)_2_·4H_2_O (2.49 g, 10 mmol, 1.0 equiv.) in ethanol (20 mL) at 25 °C. The resulting mixture was warmed to room temperature and stirred for 1 h. Ethylenediamine (2.240 g, 40 mmol, 4.0 equiv.) was then added and stirred for an additional 5 min. Next, alkynyl alcohol **8** (1.80 g, 10.0 mmol, 1.0 equiv.) was added slowly and stirred for 12 h. Then, the reaction mixture was filtered through a celite pad and concentrated under reduced pressure to obtain the crude product. The crude product was purified via silica gel chromatography (petroleum ether/ethyl acetate 10:1) to afford (*Z*)-dodec-2-en-1-ol (**11**) (1.75 g, 95% yield, *Z*:*E* > 99:1, determined via ^1^H and ^13^CNMR spectra) as a colorless oil. ^1^H NMR (500 MHz, CDCl_3_) δ 5.62–5.52 (m, 2H), 4.20 (t, *J* = 5.5 Hz, 2H), 2.07 (q, *J* = 7.5 Hz, 2H), 1.37–1.33 (m, 2H), 1.31–1.22 (m, 13H), 0.88 (t, *J* = 7.0 Hz, 3H). ^13^C NMR (125 MHz, CDCl_3_) δ 133.48, 128.41, 58.78, 32.03, 29.76, 29.71, 29.63, 29.46, 29.37, 27.58, 22.82, 14.26. HRMS (ESI, *m*/*z*): calculated for [M + Na]^+^ C_12_H_24_ONa 207.1719, found: 207.1729.

### 3.6. Synthesis of (Z)-Tridec-2-en-1-ol (***12***)

Using a similar procedure to the synthesis of **11**, tridec-2-yn-1-ol (**9**) (1.96 g, 10.0 mmol, 1.0 equiv.) afforded (*Z*)-tridec-2-en-1-ol (**12**) (1.84 g, 93% yield, *Z*:*E* > 99:1, determined via ^13^C NMR spectra) as a colorless oil. ^1^H NMR (500 MHz, CDCl_3_) δ 5.62–5.52 (m, 2H), 4.20 (t, *J* = 6.0 Hz, 2H), 2.07 (q, *J* = 7.0 Hz, 2H), 1.37–1.33 (m, 2H), 1.31–1.22 (m, 15H), 0.88 (t, *J* = 6.5 Hz, 3H).^13^C NMR (125 MHz, CDCl_3_) δ 133.48, 128.41, 58.77, 32.05, 29.76, 29.63, 29.48, 29.37, 27.58, 22.83, 14.27. HRMS (ESI, *m*/*z*): calculated for [M + Na]^+^ C_13_H_26_ONa 221.1876, found: 221.1869.

### 3.7. Synthesis of (Z)-Tetradec-2-en-1-ol (***13***)

Using a similar procedure to the synthesis of **11**, tetradec-2-yn-1-ol (**10**) (2.10 g, 10.0 mmol, 1.0 equiv.) afforded (*Z*)-tetradec-2-en-1-ol (**13**) (1.95 g, 92% yield, *Z*:*E* > 99:1, determined via ^13^C NMR spectra) as a colorless oil. ^1^H NMR (500 MHz, CDCl_3_) δ 5.62–5.52 (m, 2H), 4.20 (t, *J* = 5.0 Hz, 2H), 2.07 (q, *J* = 7.0 Hz, 2H), 1.37–1.33 (m, 2H), 1.31–1.22 (m, 17H), 0.88 (t, *J* = 6.5 Hz, 3H). ^13^C NMR (125 MHz, CDCl_3_) δ 133.48, 128.40, 58.77, 32.06, 29.81, 29.78, 29.75, 29.64, 29.49, 29.37, 27.58, 22.84, 14.27. HRMS (ESI, *m*/*z*): calculated for [M + K]^+^ C_14_H_28_OK 251.1772, found: 251.1782.

### 3.8. Synthesis of ((2S,3R)-3-Nonyloxiran-2-yl) Methanol (***14***)

Under an argon atmosphere, *L*-(+)-DIPT (2.42 g, 10.32 mmol, 1.29 equiv.) was added slowly to a suspension of powered activated 4Å molecular sieves (0.32 g) and Ti(O*i*Pr)_4_ (3.13 g, 11.0 mmol, 1.1 equiv.) in CH_2_Cl_2_ (50 mL) at −35 °C. The resulting mixture was stirred for 1 h, then (*Z*)-dodec-2-en-1-ol **11** (1.47 g, 8.0 mmol, 1.0 equiv.) was added in. After stirring for 1 h at −35 °C, *tert*-butyl hydroperoxide (TBHP) (2.8 mL, 5.5 M in decane, 15.4 mmol, 2.15 equiv.) was added slowly. The reaction mixture was warmed to −30 °C and stirred for an additional 72 h, followed by quenching with water (20 mL) at 0 °C. After stirring for 1 h, NaOH aqueous solution (40 mL, 30%) was added, and the mixture was stirred vigorously until phase separation occurred at room temperature. The organic phase was separated, and the aqueous layer was extracted with CH_2_Cl_2_ (3 × 60 mL). The combined organic layers were dried over anhydrous Na_2_SO_4_ and concentrated under reduced pressure to give crude product. The crude product was purified via silica gel chromatography (petroleum ether/diethyl ether 2:1) to afford ((2*S*,3*R*)-3-nonyloxiran-2-yl) methanol (**14**) (1.31 g, 82% yield) as a white solid. The melting point was 55.0–57.0 °C; recrystallization from petroleum ether (250 mL) provided a white solid (0.79 g, 60% yield, >99% ee, determined via the ^1^H NMR analysis of the ester derived from (*S*)-MTPACl). [α]_D_^20^ −5.55 (*c* 0.79, CHCl_3_). ^1^H NMR (500 MHz, CDCl_3_) δ 3.88–3.84 (m, 1H), 3.70–3.65 (m, 1H), 3.17–3.14 (m, 1H), 3.03 (q, *J* = 5.0 Hz, 1H), 1.86–1.73 (m, 1H), 1.58–1.27 (m, 16H), 0.88 (t, *J* = 6.5 Hz, 3H). ^13^C NMR (125 MHz, CDCl_3_) δ 61.10, 57.49, 56.96, 32.01, 29.65, 29.63, 29.56, 29.42, 28.12, 26.79, 22.81, 14.24. HRMS (ESI, *m*/*z*): calculated for [M + H]^+^ C_12_H_25_O_2_ 201.1849, found: 201.1857.

### 3.9. Synthesis of ((2S,3R)-3-Decyloxiran-2-yl) Methanol (***15***)

Using a similar procedure to the synthesis of **14**, (*Z*)-tridec-2-en-1-ol **12** (1.59 g, 8.0 mmol, 1.0 equiv.) afforded ((2*S*,3*R*)-3-decyloxiran-2-yl) methanol (**15**) (1.37 g, 80% yield) as a white solid. The melting point was 61.0–63.0 °C; recrystallization from petroleum ether (250 mL) provided a white solid (0.75 g, 55% yield, >99% ee, determined via the ^1^H NMR analysis of the ester derived from (*S*)-MTPACl). [α]_D_^20^ −3.72 (*c* 0.97, CHCl_3_). ^1^H NMR (500 MHz, CDCl_3_) δ 3.88–3.83 (m, 1H), 3.70–3.65 (m, 1H), 3.17–3.14 (m, 1H), 3.05–3.02 (m, 1H), 1.84–1.74 (m, 1H), 1.59–1.26 (m, 18H), 0.88 (t, *J* = 6.5 Hz, 3H). ^13^C NMR (125 MHz, CDCl_3_) δ 61.10, 57.49, 56.96, 32.03, 29.72, 29.67, 29.65, 29.56, 29.46, 28.12, 26.79, 22.82, 14.25. HRMS (ESI, *m*/*z*): calculated for [M + H]^+^ C_13_H_27_O_2_ 215.2006, found: 215.2016.

### 3.10. Synthesis of ((2S,3R)-3-Undecyloxiran-2-yl) Methanol (***16***)

Using a similar procedure to the synthesis of **14**, (*Z*)-tetradec-2-en-1-ol **13** (1.91 g, 8.0 mmol, 1.0 equiv.) afforded ((2*S*, 3*R*)-3-undecyloxiran-2-yl) methanol (**16**) (1.41g, 84% yield) as a white solid. The melting point was 66–68 °C; recrystallization from petroleum ether (250 mL) provided a white solid (0.88 g, 62% yield, >99% ee, determined via the ^1^H NMR analysis of the ester derived from (*S*)-MTPACl). [α]_D_^20^ −3.92 (*c* 1.63, CHCl_3_). ^1^H NMR (500 MHz, CDCl_3_) δ 3.88–3.84 (m, 1H), 3.69–3.65 (m, 1H), 3.18–3.14 (m, 1H), 3.05–3.02 (m, 1H), 2.03–2.00 (m, 1H), 1.58–1.26 (m, 20H), 0.88 (t, *J* = 6.9 Hz, 3H). ^13^C NMR (125 MHz, CDCl_3_) δ 77.41, 77.16, 76.91, 61.07, 57.48, 57.03, 32.03, 29.75, 29.74, 29.66, 29.64, 29.55, 29.46, 28.10, 26.77, 22.81, 14.23. HRMS (ESI, *m*/*z*): calculated for [M + H]^+^ C_14_H_29_O_2_ 229.2162, found: 229.2172.

### 3.11. Synthesis of (2S,3R)-2-(Oct-2-alkynyl)-3-nonylepoxyethane (***18***)

Under an argon atmosphere at −78 °C, triethylamine (0.73 g, 7.2 mmol, 3.6 equiv.) and trifluoromethanesulfonic anhydride (1.69 g, 6.0 mmol, 3.0 equiv.) were added dropwise to a vigorously stirred suspension of ((2*S*,3*R*)-3-nonyloxiran-2-yl) methanol **14** (0.40 g, 2.0 mmol, 1.0 equiv.) in anhydrous CH_2_Cl_2_ (20 mL). The suspension was allowed to warm slowly to about −60 °C and stirred. Once the solution became clear, the reaction was re-cooled to −78 °C and stirred for 30 min, then the reaction was quenched with an aqueous NH_4_Cl solution (5.0 mL). The aqueous layer was extracted with CH_2_Cl_2_ (20 mL × 3). The combined organic layers were washed with brine, dried over anhydrous Na_2_SO_4_, filtered and concentrated in a vacuum.

A solution of *n*-BuLi (2.4 mol/L in hexane, 1.5 mL, 3.6 mmol, 1.8 equiv.) was added dropwise to a solution of hept-1-yne **17** (0.38 g, 4.0 mmol, 2.0 equiv.) in anhydrous diethyl ether (20 mL) at −78 °C under an argon atmosphere. After being stirred for 10 min, a solution of crude triflate (2*S*, 3*R*)-**14** in anhydrous diethyl ether (5 mL) and anhydrous HMPA (0.8 mL) was added. After being stirred for 1 h at the same temperature, the reaction was quenched with an aqueous NH_4_Cl solution (10 mL). The aqueous layer was extracted with EtOAc (50 mL × 3). The combined organic layers were washed with brine, dried over anhydrous Na_2_SO_4_, filtered and concentrated under reduced pressure. The residue was purified via flash column chromatography on silica gel (*R*_f_ = 0.25, petroleum ether: ethyl acetate 50:1) to afford (2*S*, 3*R*)-2-(oct-2′-alkynyl)-3-nonylepoxyethane **18** (0.45 g, 80%) as a colorless oil. [α]_D_^20^ +13.85 (*c* 1.04, CHCl_3_). ^1^H NMR (500 MHz, CDCl_3_) δ 3.12–3.09 (m, 1H), 2.97–2.94 (m, 1H), 2.59–2.54 (m, 1H), 2.27–2.21 (m, 1H), 2.15 (tt, *J* = 7.2, 2.4 Hz, 2H), 1.54–1.46 (m, 4H), 1.38–1.32 (m, 4H), 1.31–1.24 (m, 14H), 0.91–0.87 (m, 6H). ^13^C NMR (125 MHz, CDCl_3_) δ 82.65, 75.02, 57.26, 55.61, 32.04, 31.21, 29.72, 29.67, 29.46, 28.75, 27.70, 26.63, 22.82, 22.36, 18.94, 18.87, 14.25, 14.13. HRMS (ESI, *m*/*z*): calculated for [M + H]^+^C_19_H_35_O 279.2682, found: 279.2692.

### 3.12. Synthesis of (2S,3R)-2-(Oct-2-alkynyl)-3-decanylepoxyethane (***19***)

Using a similar procedure to the synthesis of **18**, ((2*S*,3*R*)-3-decyloxiran-2-yl) methanol **15** (0.43 g, 2.0 mmol, 1.0 equiv.) afforded (2*S*, 3*R*)-2-(oct-2′-alkynyl)-3-decanyl epoxyethane **19** (0.46 g, 78% yield) as a colorless oil. [α]_D_^20^ +34.03 (*c* 1.54, CHCl_3_). ^1^H NMR (500 MHz, CDCl_3_) δ 3.12–3.09 (m, 1H), 2.97–2.94 (m, 1H), 2.59–2.54 (m, 1H), 2.26–2.21 (m, 1H), 2.15 (tt, *J* = 7.0, 2.0 Hz, 2H), 1.55–1.46 (m, 6H), 1.38–1.33 (m, 4H), 1.32–1.27 (m, 14H), 0.91–0.87 (m, 6H). ^13^C NMR (125 MHz, CDCl_3_) δ 82.64, 75.02, 57.25, 55.60, 32.05, 31.21, 29.75, 29.71, 29.67, 29.48, 28.75, 27.70, 26.63, 22.83, 22.36, 18.94, 18.86, 14.25, 14.13. HRMS (ESI, *m*/*z*): calculated for [M + H]^+^ C_20_H_37_O 293.2839, found: 293.2852.

### 3.13. Synthesis of (2S,3R)-2-(Oct-2-alkynyl)-3-undecylepoxyethane (***20***)

Using a similar procedure to the synthesis of **18**, ((2*S*,3*R*)-3-undecyloxiran-2-yl) methanol (0.45 g, 2.0 mmol, 1.0 equiv.) afforded (2*S*, 3*R*)-2-(oct-2′-alkynyl)-3-undecyl epoxyethane **20** (0.50 g, 81% yield) as a colorless oil. [α]_D_^20^ +21.68 (*c* 0.79, CHCl_3_). ^1^H NMR (500 MHz, CDCl_3_) δ 3.13–3.09 (m, 1H), 2.97–2.94 (m, 1H), 2.59–2.54 (m, 1H), 2.27–2.21 (m, 1H), 2.15 (tt, *J* = 7.0, 2.5 Hz, 2H), 1.55–1.46 (m, 6H), 1.37–1.32 (m, 6H), 1.31–1.26 (m, 14H), 0.91–0.8 (m, 6H). ^13^C NMR (125 MHz, CDCl_3_) δ 82.65, 75.02, 57.26, 55.61, 32.07, 31.22, 29.80, 29.78, 29.72, 29.71, 29.67, 29.50, 28.75, 27.70, 26.63, 22.84, 22.36, 18.94, 18.87, 14.26, 14.14. HRMS (ESI, *m*/*z*): calculated for [M + Na]^+^ C_21_H_38_ONa 329.2815, found: 329.2824.

### 3.14. Synthesis of (2S,3R)-2-((Z)-Oct-2-enyl)-3-nonyloxirane (***1***)

Under a hydrogen atmosphere at 0 °C, NaBH_4_ (19 mg, 0.5 mmol, 1.0 equiv.) was added to a stirred solution of Ni(OAc)_2_·4H_2_O (0.12 g, 0.5 mmol, 1.0 equiv.) in MeOH (8.0 mL) (caution: vigorous gas evolution.). The reaction mixture was allowed to warm to room temperature and then stirred for 5 min. Then, 1,2-diaminoethane (0.12g, 2.0 mmol, 4.0 equiv.) was added, and the resulting mixture was stirred for a further 5 min. A solution of (2*S*, 3*R*)-2-(oct-2′-alkynyl)-3-nonylepoxyethane **18** (0.14 g, 0.5 mmol, 1.0 equiv.) in MeOH (2.5 mL) was added. The reaction mixture stirred for 12 h. The reaction mixture was then filtered through a pad of celite, which was washed thoroughly with MeOH. The filtrate was evaporated under reduced pressure, and the residue was dissolved in Et_2_O (20 mL) and washed with H_2_O (2 × 10 mL) and brine (10 mL). Then, the filtrate was dried (MgSO_4_) and concentrated to afford the crude diene. The crude product was purified via flash chromatography (petroleum ether/ethyl acetate 50:1) to afford (2*S*,3*R*)-2-((*Z*)-oct-2′-enyl)-3-nonyl oxirane **1** (0.13 g, 90%) as a colorless oil. [α]_D_^20^ +9.45 (*c* 0.97, CHCl_3_). Lit. [α]_D_^20^ +3.06 (*c* 1.12, CH_2_Cl_2_) [3]. ^1^H NMR (500 MHz, CDCl_3_) δ 5.56–5.50 (m, 1H), 5.44–5.39 (m, 1H), 2.94–2.90 (m, 2H), 2.40–2.35 (m, 1H), 2.21–2.16 (m, 1H), 2.05 (q, *J* = 7.0 Hz, 2H), 1.55–1.47 (m, 4H), 1.38–1.33 (m, 4H), 1.32–1.27 (m, 14H), 0.90–0.87 (m, 6H). ^13^C NMR (125 MHz, CDCl_3_) δ 132.85, 123.97, 57.37, 56.71, 32.04, 31.64, 29.72, 29.67, 29.45, 29.39, 27.94, 27.56, 26.75, 26.37, 22.82, 22.70, 14.25, 14.19. HRMS (ESI, *m*/*z*): calculated for [M + H]^+^ C_19_H_37_O 281.2839, found: 281.2840.

### 3.15. Synthesis of (2S,3R)-2-((Z)-Oct-2-enyl)-3-decyloxirane (***2***)

Using a similar procedure to the synthesis of **1**, (2*S*, 3*R*)-2-(oct-2′-alkynyl)-3-decanyl epoxyethane **19** (0.15 g, 0.5 mmol, 1.0 equiv.) afforded (2*S*,3*R*)-2-((*Z*)-oct-2′-enyl)-3-decyl oxirane **2** (0.13 g, 88% yield) as a colorless oil. [α]_D_^20^ +5.31 (*c* 1.28, CHCl_3_). Lit. [α]_D_^22^ +6.96 (*c* 1.02, CHCl_3_) [5]. ^1^H NMR (500 MHz, CDCl_3_) δ 5.55–5.50 (m, 1H), 5.44–5.39 (m, 1H), 2.95–2.91 (m, 2H), 2.40–2.35 (m, 1H), 2.21–2.16 (m, 1H), 2.04 (q, *J* = 7.3 Hz, 2H), 1.55–1.41 (m, 4H), 1.39–1.34 (m, 4H), 1.30–1.27 (m, 16H), 0.88 (td, *J* = 6.9, 4.4 Hz, 6H).^13^C NMR (125 MHz, CDCl_3_) δ 132.85, 123.97, 57.37, 56.71, 32.05, 31.65, 29.75, 29.72, 29.48, 29.39, 27.94, 27.56, 26.75, 26.37, 22.83, 22.70, 14.26, 14.19. HRMS (ESI, *m*/*z*): calculated for [M + H]^+^ C_20_H_39_O 295.2995, found: 295.2981.

### 3.16. Synthesis of (2S,3R)-2-((Z)-Oct-2-enyl)-3-undecyloxirane (***3***)

Using a similar procedure to the synthesis of **1**, (2*S*, 3*R*)-2-(oct-2′-alkynyl)-3-undecyl epoxyethane **20** (0.16 g, 0.5 mmol, 1.0 equiv.) afforded (2*S*,3*R*)-2-((*Z*)-oct-2′-enyl)-3-undecyl oxirane **3** (0.14 g, 92% yield) as a colorless oil. [α]_D_^20^ +5.95 (*c* 0.74, CHCl_3_). Lit. [α]_D_^22^ +6.55 (*c* 1.15, CHCl_3_) [5]. ^1^H NMR (500 MHz, CDCl_3_) δ 5.56–5.50 (m, 1H), 5.44–5.38 (m, 1H), 2.94–2.92 (m, 2H), 2.40–2.35 (m, 1H), 2.22–2.16 (m, 1H), 2.04 (q, *J* = 7.3 Hz, 2H), 1.56–1.49 (m, 2H), 1.43–1.34 (m, 6H), 1.30–1.26 (m, 18H), 0.90–0.87 (m, 6H). ^13^C NMR (125 MHz, CDCl_3_) δ 132.85, 123.97, 57.38, 56.72, 32.07, 31.65, 29.79, 29.78, 29.72, 29.49, 29.40, 27.94, 27.57, 26.76, 26.38, 22.84, 22.70, 14.26, 14.20. HRMS (ESI, *m*/*z*): calculated for [M + H]^+^ C_21_H_41_O 309.3152, found: 309.3162.

## 4. Conclusions

In summary, we have developed an efficient and novel asymmetric synthesis of the sex pheromone of the elm spanworm (**1**) and the painted apple moth (**2** and **3**). The central components of our strategy involved the alkylation of acetylide ion to connect chiral epoxy triflate with alkyne, Sharpless asymmetric epoxidation to construct the stereocenters, and Brown’s P2-Ni reduction to provide a *cis* alkene. Compared to the reported procedures, this approach has the advantages of cheaper starting materials, a shorter synthetic route, higher total yield and higher enantiopurity. Our research would be beneficial for the control of *Ennomus subsignaria* (Hübner) and *Teia anartoides* (Walker).

## Data Availability

The data presented in this article are available in the Appendix A.

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
