# Peer review of "Asymmetric Synthesis of Three Alkenyl Epoxides: Crafting the Sex Pheromones of the Elm Spanworm and the Painted Apple Moth"

_molecules, 2024, doi:10.3390/molecules29092136_

Round 1
Reviewer 1 Report
Comments and Suggestions for Authors
This is an asymmetric synthesis of (2S,3R)-2-((Z)-oct-2-enyl)-3-nonyloxirane and its homologs, which are pheromone components of lepidopteran pests, using the Sharpless asymmetric epoxidation. The yields and the enantiomeric excess values of the products are pretty good. Thus I think this report is valuable as a piece of works of bioactive molecule syntheses. However, I found several issues and relatively many incorrect descriptions that should be addressed before considering publication.
One of my major concerns is the process of recrystallization of chiral epoxy alcohols to improve ee values to >99%. Because ee values of products in the Sharpless asymmetric epoxidation are usually not perfect, the present recrystallization is worth discussed. I would like to request the authors to provide more descriptions about plausible reasons why their recrystallization can improve ee values of epoxy alcohols.
Other editorial points are as follows;
L22
forest -> forestry
L24
ash (Franxinus)) -> ash (Franxinus)
L27
larve -> larvae
L38 and others
(2S,3R)-2-((Z)-oct-2’-enyl)-3-nonyl oxirane -> (2S,3R)-2-((Z)-oct-2-enyl)-3-nonyloxirane [no need of “ ’ ” on “2”, and remove “_(space)” between “nonyl” and “oxirane” ]
L43
Sayed’s team -> El-Sayed’s team
L46
theses -> these
L77 and others
(2S, 3R) -2- (oct-2 '- alkynyl) -3-nonylepoxyethane -> (2S,3R)-2-(oct-2-ynyl)-3-nonylepoxyethane
L78 and others
propargyl -> alkynyl
L95
L-(+)-diethyl tartrate -> “L” should not be in italic.
L389,391
There are no journal information.
Comments on the Quality of English LanguageI found several misspellings and grammatical errors.
Reviewer 2 Report
Comments and Suggestions for Authors Line 87: To be rewritten: "Our synthesis commenced with the synthesis of..."Line 91: "catalyst of Brown’s" to be corrected to Brown's catalyst
Line 107: 18 is not an alcohol, please correct the phrase, as well as the 19 and 20
Comments on the Quality of English Language
Minor corrections of the English are required
Reviewer 3 Report
Comments and Suggestions for Authors
The manuscript submitted by Shan, and co-workers describes the asymmetric synthesis of three alkenyl epoxides of sex pheromones with the key steps of 1) the alkylation of acetylide ion, 2) Sharpless asymmetric epoxidation, and 3) Brown’s P2-Ni reduction. The synthesis mostly followed Mori’s asymmetric synthesis of compound 3, as described in reference 15, which included 1) the alkylation of acetylide ions, 2) Sharpless asymmetric epoxidation, 3) reduction with Lindlar’s catalyst, and 4) recrystallization of 2,3-epoxyalcohol 16 to improve the enantiomeric ratio. However, the improved overall yield and application in the synthesis of three alkenyl epoxides with 99% ee could be valuable for developing these sex pheromones as pest management tools in the future. Therefore, I recommend that the manuscript be published in Molecules after addressing the following minor remarks.
1) L11: As mentioned above, the synthetic pathway is almost the same as that of Mori (ref. 15), the words “novel and” should be removed.
2) L14: The total yields of 38%, 32%, and 38% are incorrect and should be corrected to 31%, 26%, and 32%, respectively.
3) Miscalculation of the total yields caused by the unclear recrystallization yields of compounds 14–16 in Scheme 2 (for details, see LL207–229 on page 6). Therefore, the recrystallization yield of compounds 14–16 (60%, 55%, and 62%) in Scheme 2 should be changed to “49% yield from 11”, “44% yield from 12”, and “52% yield from 13”, respectively.
4) L91: In addition to reference 17, the original report from Brown et al. should be cited (J. Org. Chem. 1973, 38, 2226).
5) Scheme 3: I believe the final steps described in Scheme 3 are the most important parts of this manuscript, because the conversion of 14–16 to 1–3 was achieved in 75%–69% yields, whereas that of Mori’s synthesis resulted in 41.7% yield. I suggest that these improvements should be described in the main text with the advantage of their alkynylation/reduction approach, rather than Mori’s addition reaction of vinyl cuprate (ref. 15).
6) LL172,180,188: The authors explained that the Z/E ratio was determined by 13C NMR spectra. However, the integral ratio of 13C NMR spectra is not reliable because of the poor S/N ratio. Thus, the Z/E ratio should be determined using other methods, such as 1H NMR and HPLC analyses.
Other points:
1) L25: Remove an extra “)” from “(Fraxinus))”.
2) L27: Remove period from “New Zealand.”.
3) L43: Insert a space before and after “Notably,”.
4) L45: The year “1999” should be “1986” of ref. 10.
5) L49: Use small capital for “D” of “D-xylose” and “X” of xylose should be a lowercase letter.
6) L50: Insert a space between “1986” and “[10]”.
7) L80: The term “enol” refers to vinylic alcohol, is inappropriate here, and should be corrected to “allylic alcohol”.
8) Scheme 1: One carbon atom is missing from the structure of 1-bromononane (4).
9) L87: Remove duplicated “chiral”.
10) L95: The authors used “L-(+)-diethyl tartrate” in the main text, but diisopropyl tartrate (DIPT) was used in Scheme 2 and the Materials and Methods section. Correct to either one.
11) Scheme 2: “NiAc2” should be corrected to “Ni(OAc)2”.
12) Scheme 2: “i” of “OiPr” should be italicized.
13) LL291, 302, 312: Insert solvents used for the column.
14) L315: The coupling pattern of “mkki” should be “m”.
15) Ref. 1: Remove “J. T. c. o. p.; I, o. s.”.
16) Ref. 8: Remove “J. P. m. s.”.
17) Ref. 9: Remove “J. P. i. l. r.”.
18) Ref. 12: The journal name should be “J. Chem. Soc., Perkin Trans. 1”.
19) Ref. 19: Substitute “J. T. A.” to “Wang, M.; Wang, M; Bian, Q”
20) Refs. 19 and 20: Insert the journal name “Tetrahedron: Asymmetry”.
21) Ref. 20: Remove “J. T. A.”.
22) Ref. 21: Remove the space between “Chem” and “.”.
